# Deep Symmetry Networks

**Robert Gens**      **Pedro Domingos**
Department of Computer Science and Engineering
University of Washington
Seattle, WA 98195-2350, U.S.A.
`{rcg,pedrod}@cs.washington.edu`

## Abstract

The chief difficulty in object recognition is that objects' classes are obscured by a large number of extraneous sources of variability, such as pose and part deformation. These sources of variation can be represented by symmetry groups, sets of composable transformations that preserve object identity. Convolutional neural networks (convnets) achieve a degree of translational invariance by computing feature maps over the translation group, but cannot handle other groups. As a result, these groups' effects have to be approximated by small translations, which often requires augmenting datasets and leads to high sample complexity. In this paper, we introduce deep symmetry networks (symnets), a generalization of convnets that forms feature maps over arbitrary symmetry groups. Symnets use kernel-based interpolation to tractably tie parameters and pool over symmetry spaces of any dimension. Like convnets, they are trained with backpropagation. The composition of feature transformations through the layers of a symnet provides a new approach to deep learning. Experiments on NORB and MNIST-rot show that symnets over the affine group greatly reduce sample complexity relative to convnets by better capturing the symmetries in the data.

## 1   Introduction

Object recognition is a central problem in vision. What makes it challenging are all the nuisance factors such as pose, lighting, part deformation, and occlusion. It has been shown that if we could remove these factors, recognition would be much easier [2, 17]. Convolutional neural networks (convnets), the current state-of-the-art method for object recognition, capture only one type of invariance (translation); the rest have to be approximated via it and standard features. In practice, the best networks require enormous datasets which are further expanded by affine transformations [7, 13] yet are sensitive to imperceptible image perturbations [23]. We propose deep symmetry networks, a generalization of convnets based on symmetry group theory [20] that makes it possible to capture a broad variety of invariances, and correspondingly improves generalization.

A symmetry group is a set of transformations that preserve the identity of an object and obey the group axioms. Most of the visual nuisance factors are symmetry groups themselves, and by incorporating them into our model we are able to reduce the sample complexity of learning from data transformed by these groups. Deep symmetry networks (symnets) form feature maps over any symmetry group, rather than just the translation group. A feature map in a deep symmetry network is defined analogously to convnets as a filter that is applied at all points in the symmetry space. Each layer in our general architecture is constructed by applying every symmetry in the group to the input, computing features on the transformed input, and pooling over neighborhoods. The entire architecture is then trained by backpropagation. In this paper, we instantiate the architecture with the affine group, resulting in deep affine networks. In addition to translation, the affine group includes rotation, scaling and shear. The affine group of the two-dimensional plane is six-dimensional (i.e., an affine transformation can be represented by a point in 6D affine space). The key challenge with

extending convnets to affine spaces is that it is intractable to explicitly represent and compute with a high-dimensional feature map. We address this by approximating the map using kernel functions, which not only interpolate but also control pooling in the feature maps. Compared to convnets, this architecture substantially reduces sample complexity on image datasets involving 2D and 3D transformations.

We share with other researchers the hypothesis that explanatory factors cannot be disentangled unless they are represented in an appropriate symmetry space [4, 11]. Our adaptation of a representation to work in symmetry space is similar in some respects to the use of tangent distance in nearest-neighbor classifiers [22]. Symnets, however, are deep networks that compute features in symmetry space at every level. Whereas the tangent distance approximation is only locally accurate, symnet feature maps can represent large displacements in symmetry space. There are other deep networks that reinterpret the invariance of convolutional networks. Scattering networks [6] are cascades of wavelet decompositions designed to be invariant to particular Lie groups, where translation and rotation invariance have been demonstrated so far. The M-theory of Anselmi et al. [2] constructs features invariant to a symmetry group by using statistics of dot products with group orbits. We differ from these networks in that we model multiple symmetries jointly in each layer, we do not completely pool out a symmetry, and we discriminatively train our entire architecture. The first two differences are important because objects and their subparts may have relative flexibility but not total invariance along certain dimensions of symmetry space. For example, a leg of a person can be seen in some but not all combinations of rotation and scale relative to the torso. Without discriminative training, scattering networks and M-theory are limited to representing features whose invariances may be inappropriate for a target concept because they are fixed ahead of time, either by the wavelet hierarchy of the former or unsupervised training of the latter. The discriminative training of symnets yields features with task-oriented invariance to their sub-features. In the context of digit recognition this might mean learning the concept of a '0' with more rotation invariance than a '6', which would incur loss if it had positive weights in the region of symmetry space where a '9' would also fire.

Much of the vision literature is devoted to features that reduce or remove the effects of certain symmetry groups, e.g., [18, 17]. Each feature by itself is not discriminative for object recognition, so structure is modeled separately, usually with a representation that does not generalize to novel viewpoints (e.g., bags-of-features) or with a rigid alignment algorithm that cannot represent uncertainty over geometry (e.g. [9, 19]). Compared to symnets, these features are not learned, have invariance limited to a small set of symmetries, and destroy information that could be used to model object sub-structure. Like deformable part models [10], symnets can model and penalize relative transformations that compose up the hierarchy, but can also capture additional symmetries.

Symmetry group theory has made a limited number of appearances in machine learning [8]. A few applications are discussed by Kondor [12], and they are also used in determinantal point processes [14]. Methods for learning transformations from examples [24, 11] could potentially benefit from being embedded in a deep symmetry network. Symmetries in graphical models [21] lead to effective lifted probabilistic inference algorithms. Deep symmetry networks may be applicable to these and other areas.

In this paper, we first review symmetry group theory and its relation to sample complexity. We then describe symnets and their affine instance, and develop new methods to scale to high-dimensional symmetry spaces. Experiments on NORB and MNIST-rot show that affine symnets can reduce by a large factor the amount of data required to achieve a given accuracy level.

## 2   Symmetry Group Theory

A symmetry of an object is a transformation that leaves certain properties of that object intact [20]. A group is a set $\mathcal{S}$ with an operator $*$ on it with the four properties of closure, associativity, an identity element, and an inverse element. A symmetry group is a type of group where the group elements are functions and the operator is function composition. A simple geometric example is the symmetry group of a square, which consists of four reflections and $\{0, 1, 2, 3\}$ multiples of 90-degree rotations. These transformations can be composed together to yield one of the original eight symmetries. The identity element is the 0-degree rotation. Each symmetry has a corresponding inverse element. Composition of these symmetries is associative.

Lie groups are continuous symmetry groups whose elements form a smooth differentiable manifold. For example, the symmetries of a circle include reflections and rotations about the center. The affine group is a set of transformations that preserves collinearity and parallel lines. The Euclidean group is a subgroup of the affine group that preserves distances, and includes the set of rigid body motions (translations and rotations) in three-dimensional space.

The elements of a symmetry group can be represented as matrices. In this form, function composition can be performed via matrix multiplication. The transformation $\mathbf{P}$ followed by $\mathbf{Q}$ (also denoted $\mathbf{Q} \circ \mathbf{P}$) is computed as $\mathbf{R} = \mathbf{QP}$. In this paper we treat the transformation matrix $\mathbf{P}$ as a point in $D$-dimensional space, where $D$ depends on the particular representation of the symmetry group (e.g., $D = 6$ for affine transformations in the plane).

A generating set of a group is a subset of the group such that any group element can be expressed through combinations of generating set elements and their inverses. For example, a generating set of the translation symmetry group is $\{x \rightarrow x + \epsilon, y \rightarrow y + \epsilon\}$ for infinitesimal $\epsilon$. We define the *k-neighborhood* of element $f$ in group $\mathcal{S}$ under generating set $\mathcal{G}$ as the subset of $\mathcal{S}$ that can be expressed as $f$ composed with elements of $\mathcal{G}$ or their inverses at most $k$ times. With the previous example, the $k$-neighborhood of a translation vector $f$ would take the shape of a diamond centered at $f$ in the *xy*-plane.

The orbit of an object $x$ is the set of objects obtained by applying each element of a symmetry group to $x$. Formally, a symmetry group $\mathcal{S}$ acting on a set of objects $X$ defines an orbit for each $x \in X$: $O_x = \{s * x : s \in \mathcal{S}\}$. For example, the orbit of an image $I(u)$ whose points are transformed by the rotation symmetry group $s * I(u) = I(s^{-1} * u)$ is the set of images resulting from all rotations of that image. If two orbits share an element, they are the same orbit. In this way, a symmetry group $\mathcal{S}$ partitions the set of objects into unique orbits $X = \bigcup_a O_a$. If a data distribution $\mathcal{D}(x, y)$ has the property that all the elements of an orbit share the same label $y$, $\mathcal{S}$ imposes a constraint on the hypothesis class of a learner, effectively lowering its VC-dimension and sample complexity [1].

## 3  Deep Symmetry Networks

Deep symmetry networks represent rich compositional structure that incorporates invariance to high-dimensional symmetries. The ideas behind these networks are applicable to any symmetry group, be it rigid-body transformations in 3D or permutation groups over strings. The architecture of a symnet consists of several layers of feature maps. Like convnets, these feature maps benefit from weight tying and pooling, and the whole network is trained with backpropagation. The maps and the filters they apply are in the dimension $D$ of the chosen symmetry group $\mathcal{S}$.

A deep symmetry network has L layers $l \in \{1, ..., L\}$ each with $I_l$ features and corresponding feature maps. A feature is the dot-product of a set of weights with a corresponding set of values from a local region of a lower layer followed by a nonlinearity. A feature map represents the application of a filter at all points in symmetry space. A feature at point $\mathbf{P}$ is computed from the feature maps of the lower layer at points in the $k$-neighborhood of $\mathbf{P}$. As $\mathbf{P}$ moves in the symmetry space of a feature map, so does its neighborhood of inputs in the lower layer. Feature map $i$ of layer $l$ is denoted $M[l, i] : \mathbb{R}^D \rightarrow \mathbb{R}$, a scalar function of the $D$-dimensional symmetry space. Given a generating set $\mathcal{G} \subset \mathcal{S}$, the points in the $k$-neighborhood of the identity element are stored in an array $\mathbf{T}[\,]$. Each filter $i$ of layer $l$ defines a weight vector $\mathbf{w}[l, i, j]$ for each point $\mathbf{T}[j]$ in the $k$-neighborhood. The vector $\mathbf{w}[l, i, j]$ is the size of $I_{l-1}$, the number of features in the underlying layer. For example, a feature in an affine symnet that detects a person would have positive weight for an arm sub-feature in the region of the $k$-neighborhood that would transform the arm relative to the person (e.g., smaller, rotated, and translated relative to the torso). The value of feature map $i$ in layer $l$ at point $\mathbf{P}$ is the dot-product of weights and underlying feature values in the neighborhood of $\mathbf{P}$ followed by a nonlinearity:

$$M[l, i](\mathbf{P}) = \sigma\left(v(\mathbf{P}, l, i)\right) \tag{1}$$

$$v(\mathbf{P}, l, i) = \sum_j^{|\mathbf{T}|} \mathbf{w}[l, i, j] \cdot \mathbf{x}(\mathbf{P} \circ \mathbf{T}[j]) \tag{2}$$

$$\mathbf{x}(\mathbf{P}') = \left\langle \begin{matrix} S(M[l-1, 0])(\mathbf{P}') \\ \dots \\ S(M[l-1, I_{l-1}])(\mathbf{P}') \end{matrix} \right\rangle \tag{3}$$

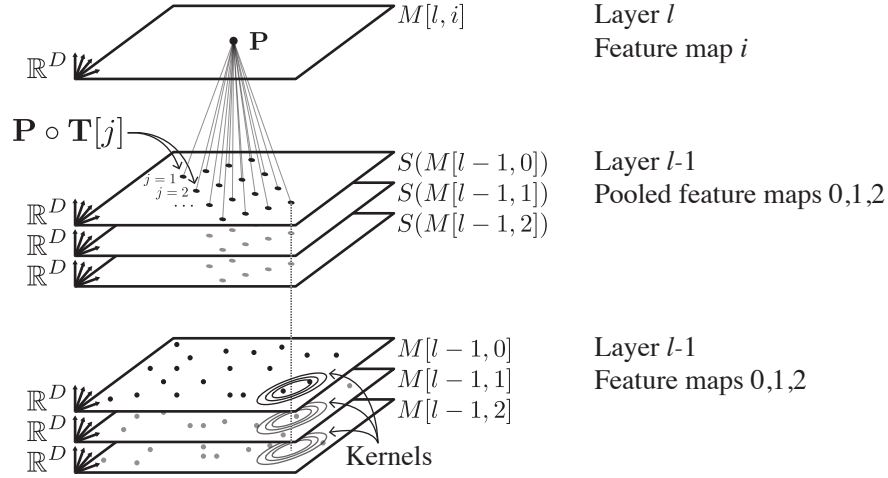

Figure 1: The evaluation of point $\mathbf{P}$ in map $M[l, i]$. The elements of the $k$-neighborhood of $\mathbf{P}$ are computed $\mathbf{P} \circ \mathbf{T}[j]$. Each point in the neighborhood is evaluated in the pooled feature maps of the lower layer $l - 1$. The pooled maps are computed with kernels on the underlying feature maps. The dashed line intersects the points in the pooled map whose values form $\mathbf{x}(\mathbf{P} \circ \mathbf{T}[j])$ in Equation 3; it also intersects the contours of kernels used to compute those pooled values. The value of the feature is the sum of the dot-products $\mathbf{w}[l, i, j] \cdot \mathbf{x}(\mathbf{P} \circ \mathbf{T}[j])$ over all $j$, followed by a nonlinearity.

where $\sigma$ is the nonlinearity (e.g., $\tanh(x)$ or $\max(x, 0)$), $v(\mathbf{P}, l, i)$ is the dot product, $\mathbf{P} \circ \mathbf{T}[j]$ represents element $j$ in the $k$-neighborhood of $\mathbf{P}$, and $\mathbf{x}(\mathbf{P}')$ is the vector of values from the underlying pooled maps at point $\mathbf{P}'$. This definition is a generalization of feature maps in convnets[1]. Similarly, the same filter weights $\mathbf{w}[l, i, j]$ are tied across all points $\mathbf{P}$ in feature map $M[l, i]$. The evaluation of a point in a feature map is visualized in Figure 1.

Feature maps $M[l, i]$ are pooled via kernel convolution to become $S(M[l, i])$. In the case of sum-pooling, $S(M[l, i])(\mathbf{P}) = \int M[l, i](\mathbf{P} - \mathbf{Q})K(\mathbf{Q}) \, d\mathbf{Q}$; for max-pooling, $S(M[l, i])(\mathbf{P}) = \max_Q M[l, i](\mathbf{P} - \mathbf{Q})K(\mathbf{Q})$. The kernel $K(\mathbf{Q})$ is also a scalar function of the $D$-dimensional symmetry space. In the previous example of a person feature, the arm feature map could be pooled over a wide range of rotations but narrow range of translations and scales so that the person feature allows for moveable but not unrealistic arms. Each filter can specify the kernels it uses to pool lower layers, but for the sake of brevity and analogy to convnets we assume that the feature maps of a layer are pooled by the same kernel. Note that convnets discretize these operations, subsample the pooled map, and use a uniform kernel $K(\mathbf{Q}) = \mathbb{1}\{\|\mathbf{Q}\|_\infty < r\}$.

As with convnets, the values of points in a symnet feature map are used by higher symnet layers, layers of fully connected hidden units, and ultimately softmax classification. Hidden units take the familiar form $\mathbf{o} = \sigma(\mathbf{W}\mathbf{x} + \mathbf{b})$, with input $\mathbf{x}$, output $\mathbf{o}$, weight matrix $\mathbf{W}$, and bias $\mathbf{b}$. The log-loss of the softmax $\mathcal{L}$ on an instance is $-\mathbf{w_i} \cdot \mathbf{x} - b_i + \log\left(\sum_c \exp\left(\mathbf{w_c} \cdot \mathbf{x} + b_c\right)\right)$, where $Y = i$ is the true label, $\mathbf{w_c}$ and $b_c$ are the weight vector and bias for class $c$, and the summation is over the classes. The input image is treated as a feature map (or maps, if color or stereo) with values in the translation symmetry space.

Deep symmetry networks are trained with backpropagation and are amenable to the same best practices as convnets. Though feature maps are defined as continuous, in practice the maps and their gradients are evaluated on a finite set of points $\mathbf{P} \in M[l, i]$. We provide the partial derivative of the loss $\mathcal{L}$ with respect to a weight vector.

$$\frac{\partial \mathcal{L}}{\partial \mathbf{w}[l, i, j]} = \sum_{\mathbf{P} \in M[l, i]} \frac{\partial \mathcal{L}}{\partial M[l, i](\mathbf{P})} \frac{\partial M[l, i](\mathbf{P})}{\partial \mathbf{w}[l, i, j]} \tag{4}$$

$$\frac{\partial M[l, i](\mathbf{P})}{\partial \mathbf{w}[l, i, j]} = \sigma'\left(v(\mathbf{P}, l, i)\right) \mathbf{x}(\mathbf{P} \circ \mathbf{T}[j]) \tag{5}$$

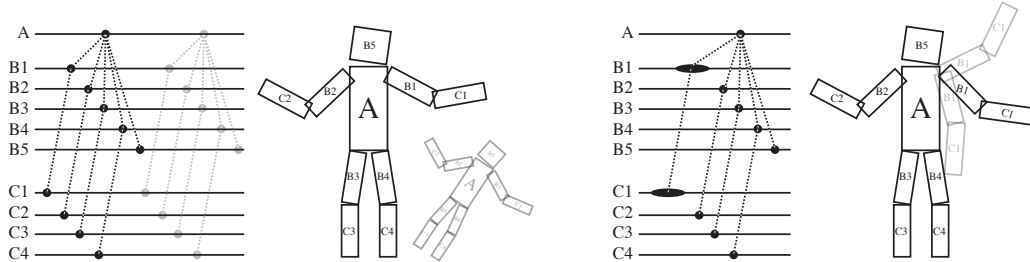

Figure 2: The feature hierarchy of a three-layer deep affine net is visualized with and without pooling. From top to bottom, the layers (A,B,C) contain one, five, and four feature maps, each corresponding to a labeled part of the cartoon figure. Each horizontal line represents a six-dimensional affine feature map, and bold circles denote six-dimensional points in the map. The dashed lines represent the affine transformation from a feature to the location of one of its filter points. For clarity, only a subset of filter points are shown. **Left:** Without pooling, the hierarchy represents a rigid affine transformation among all maps. Another point on feature map A is visualized in grey. **Right:** Feature maps B1 and C1 are pooled with a kernel that gives those features flexibility in rotation.

The partial derivative of the loss $\mathcal{L}$ with respect to the value of a point in a lower layer is

$$\frac{\partial \mathcal{L}}{\partial M[l-1,i](\mathbf{P})} = \sum_{i'}^{I_l} \sum_{\mathbf{P}' \in M[l,i']} \frac{\partial \mathcal{L}}{\partial M[l,i'](\mathbf{P}')} \frac{\partial M[l,i'](\mathbf{P}')}{\partial M[l-1,i](\mathbf{P})} \tag{6}$$

$$\frac{\partial M[l,i'](\mathbf{P}')}{\partial M[l-1,i](\mathbf{P})} = \sigma'\left(v(\mathbf{P}',\mathbf{l},\mathbf{i}')\right) \sum_j^{|\mathbf{T}|} \mathbf{w}[l,i',j][i] \frac{\partial S(M[l-1,i])(\mathbf{P}' \circ \mathbf{T}[j])}{\partial M[l-1,i](\mathbf{P})} \tag{7}$$

where the gradient of the pooled feature map $\frac{\partial S(M[l,i])(\mathbf{P})}{\partial M[l,i](\mathbf{Q})}$ equals $K(\mathbf{P} - \mathbf{Q})$ for sum-pooling.

None of this treatment depends explicitly on the dimensionality of the space except for the kernel and transformation composition which have polynomial dependence on $D$. In the next section we apply this architecture to the affine group in 2D, but it could also be applied to the affine group in 3D or any other symmetry group.

## 4 Deep Affine Networks

We instantiate a deep symmetry network with the affine symmetry group in the plane. The affine symmetry group contains transformations capable of rotating, scaling, shearing, and translating two-dimensional points. The transformation is described by six coordinates:

$$\begin{bmatrix} x' \\ y' \end{bmatrix} = \begin{bmatrix} a & b \\ c & d \end{bmatrix} \begin{bmatrix} x \\ y \end{bmatrix} + \begin{bmatrix} e \\ f \end{bmatrix}$$

This means that each of the feature maps $M[l,i]$ and elements $\mathbf{T}[j]$ of the $k$-neighborhood is represented in six dimensions. The identity transformation is $a = d = 1, b = c = e = f = 0$. The generating set of the affine symmetry group contains six elements, each of which is obtained by adding $\epsilon$ to one of the six coordinates in the identity transform. This generating set is visualized in Figure 3.

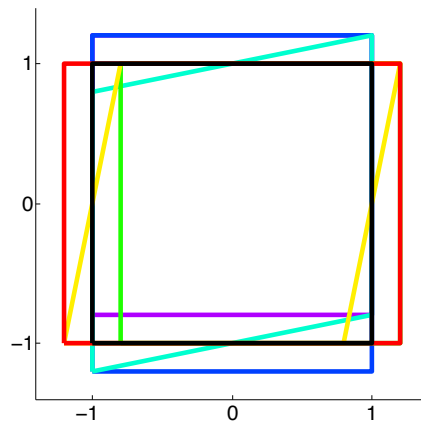

Figure 3: The six transformations in the generating set of the affine group applied to a square (exaggerated $\epsilon = 0.2$, identity is black square).

A deep affine network can represent a rich part hierarchy where each weight of a feature modulates the response to a subpart at a point in the affine neighborhood. The geometry of a deep affine network is best understood by tracing a point on a feature map through its filter point transforms into lower layers. Figure 2 visualizes this structure without and with pooling on the left and right sides of the diagram, respectively. Without pooling, the feature hierarchy defines a rigid affine relationship between the point of evaluation on a map and the location of its sub-features. In contrast, a pooled value on a sub-feature map is computed from a neighborhood defined by the kernel of points in affine space; this can represent model flexibility along certain dimensions of affine space.

## 5 Scaling to High-Dimensional Symmetry Spaces

It would be intractable to explicitly represent the high-dimensional feature maps of symnets. Even a subsampled grid becomes unwieldy at modest dimensions (e.g., a grid in affine space with ten steps per axis has $10^6$ points). Instead, each feature map is evaluated at $N$ control points. The control points are local maxima of the feature in symmetry space, found by Gauss-Newton optimization, each initialized from a prior. This can be seen as a form of non-maximum suppression. Since the goal is recognition, there is no need to approximate the many points in symmetry space where the feature is not present. The map is then interpolated with kernel functions; the shape of the function also controls pooling.

### 5.1 Transformation Optimization

Convnets max-pool a neighborhood of translation space by exhaustive evaluation of feature locations. There are a number of algorithms that solve for a maximal feature location in symmetry space but they are not efficient when the feature weights are frequently adjusted [9, 19]. We adopt an iterative approach that dovetails with the definition of our features.

If a symnet is based on a Lie group, gradient based optimization can be used to find a point $\mathbf{P}^*$ that locally maximizes the feature value (Equation 1) initialized at point $\mathbf{P}$. In our experiments with deep affine nets, we follow the forward compositional (FC) warp [3] to align filters with the image. An extension of Lucas-Kanade, FC solves for an image alignment. We adapt this procedure to our filters and weight vectors: $\min_{\Delta \mathbf{P}} \sum_j^{|\mathbf{T}|} \|\mathbf{w}[l,i,j] - \mathbf{x}(\mathbf{P} \circ \Delta \mathbf{P} \circ \mathbf{T}[j])\|^2$. We run an FC alignment for each of the $N$ control points in feature map $M[l,i]$, each initialized from a prior. Assuming $\sum_j^{|\mathbf{T}|} \|\mathbf{x}(\mathbf{P} \circ \Delta \mathbf{P} \circ \mathbf{T}[j])\|^2$ is constant, this procedure locally maximizes the dot product between the filter and the map in Equation 2. Each iteration of FC takes a Gauss-Newton step to solve for a transformation of the neighborhood of the feature in the underlying map $\Delta \mathbf{P}$, which is then composed with the control point: $\mathbf{P} \leftarrow \mathbf{P} \circ \Delta \mathbf{P}$.

### 5.2 Kernels

Given a set of $N$ local optima $O^* = \{(\mathbf{P}_1, v_1), \ldots, (\mathbf{P}_N, v_N)\}$ in $D$-dimensional feature map $M[l,i]$, we use kernel-based interpolation to compute a pooled map $S(M[l,i])$. The kernel performs three functions: penalizing relative locations of sub-features in symmetry space (*cf.* [10]), interpolating the map, and pooling a region of the map. These roles could be split into separate filter-specific kernels that are then convolved appropriately. The choice of these kernels will vary with the application. In our experiments, we lump these functions into a single kernel for a layer. We use a Gaussian kernel $K(\mathbf{Q}) = e^{-\mathbf{q}^T \Sigma^{-1} \mathbf{q}}$ where $\mathbf{q}$ is the $D$-dimensional vector representation of $\mathbf{Q}$ and the $D \times D$ covariance matrix $\Sigma$ controls the shape and extent of the kernel. Several instances of this kernel are shown in Figure 4. Max-pooling produced the best results on our tests.

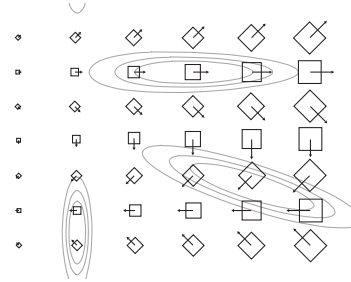

Figure 4: Contours of three 6D Gaussian kernels visualized on a surface in affine space. Points are visualized by an oriented square transformed by the affine transformation at that point. Each kernel has a different covariance matrix $\Sigma$.

## 6 Experiments

In our experiments we test the hypothesis that a deep network with access to a larger symmetry group will generalize better from fewer examples, provided those symmetries are present in the data. In particular, theory suggests that a symnet will have better sample complexity than another classifier on a dataset if it is based on a symmetry group that generates variations present in that dataset [1]. We compare deep affine symnets to convnets on the MNIST-rot and NORB image classification datasets, which finely sample their respective symmetry spaces such that learning curves measure the amount of augmentation that would be required to achieve similar performance. On both datasets affine symnets achieve a substantial reduction in sample complexity. This is particularly remarkable on NORB because its images are generated by a symmetry space in 3D. Symnet running time was within an order of magnitude of convnets, and could be greatly optimized.

### 6.1 MNIST-rot

MNIST-rot [15] consists of 28x28 pixel greyscale images: $10^4$ for training, $2 \times 10^3$ for validation, and $5 \times 10^4$ for testing. The images are sampled from the MNIST digit recognition dataset and each

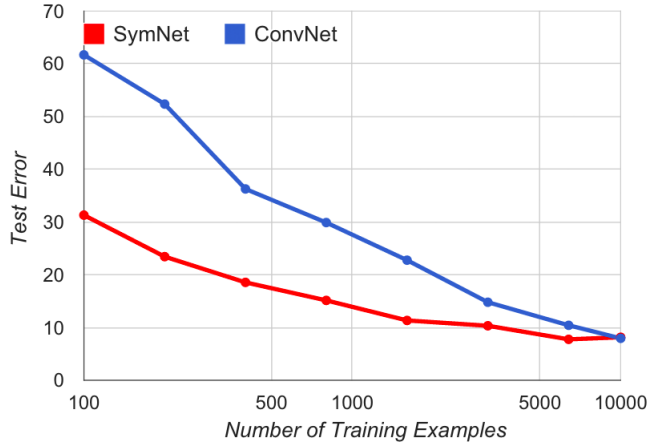

Figure 5: Impact of training set size on MNIST-rot test performance for architectures that use either one convolutional layer or one affine symnet layer.

is rotated by a random angle in the uniform distribution $[0, 2\pi]$. With transformations that apply to the whole image, MNIST-rot is a good testbed for comparing the performance of a single affine layer to a single convnet layer.

We modified the Theano [5] implementation of convolutional networks so that the network consisted of a single layer of convolution and maxpooling followed by a hidden layer of 500 units and then softmax classification. The affine net layer was directly substituted for the convolutional layer. The control points of the affine net were initialized at uniformly random positions with rotations oriented around the image center, and each control point was locally optimized with four iterations of Gauss-Newton updates. The filter points of the affine net were arranged in a square grid. Both the affine net and the convnet compute a dot-product and use the sigmoid nonlinearity. Both networks were trained with 50 epochs of mini-batch gradient descent with momentum, and test results are reported on the network with lowest error on the validation set[2]. The convnet did best with small $5 \times 5$ filters and the symnet with large $20 \times 20$ filters. This is not surprising because the convnet must approximate the large rotations of the dataset with translations of small patches. The affine net can pool directly in this space of rotations with large filters.

Learning curves for the two networks are presented in Figure 5. We observe that the affine symnet roughly halves the error of the convnet. With small sample sizes, the symnet achieves an accuracy for which the convnet requires about eight times as many samples.

## 6.2  NORB

MNIST-rot is a synthetic dataset with symmetries that are not necessarily representative of real images. The NORB dataset [16] contains $2 \times 108 \times 108$ pixel stereoscopic images of 50 toys in five categories: quadrupeds, human figures, airplanes, trucks, and cars. Five of the ten instances of each category are reserved for the test set. Each toy is photographed on a turntable from an exhaustive set of angles and lighting conditions. Each image is then perturbed by a random translation shift, planar rotation, luminance change, contrast change, scaling, distractor object, and natural image background. A sixth blank category containing just the distractor and background is also used. As in other papers, we downsample the images to $2 \times 48 \times 48$. To compensate for the effect of distractors in smaller training sets, we also train and test on a version of the dataset that is centrally-cropped to $2 \times 24 \times 24$. We report results for whichever version had lower validation error. In our experiments we train on a variable subset of the first training fold, using the first $2 \times 10^3$ images of the second fold for validation. Our results use both of the testing folds.

We compare architectures that use two convolutional layers or two affine ones, which performed better than single-layer ones. As with the MNIST-rot experiments, the symnet and convnet layers are followed by a layer of 500 hidden units and softmax classification. The symnet control points in the first layer were arranged in three concentric rings in translation space, with 8 points spaced across rotation (200 total points). Control points in the second layer were fixed at the center of translation

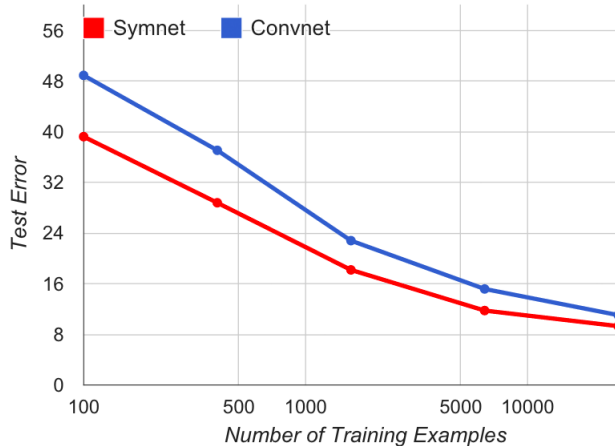

Figure 6: Impact of training set size on NORB test performance for architectures with two convolutional or affine symnet layers followed by a fully connected layer and then softmax classification.

space arranged over 8 rotations and up to 2 vertical scalings (16 total points) to approximate the effects of elevation change. Control points were not iteratively optimized due to the small size of object parts in downsampled images. The filter points of the first layer of the affine net were arranged in a square grid. The second layer filter points were arranged in a circle in translation space at a 3 or 4 pixel radius, with 8 filter points evenly spaced across rotation at each translation. We report the test results of the networks with lowest validation error on a range of hyperparameters[3].

The learning curves for convnets and affine symnets are shown in Figure 6. Even though the primary variability in NORB is due to rigid 3D transformations, we find that our affine networks still have an advantage over convnets. A 3D rotation can be locally approximated with 2D scales, shears, and rotations. The affine net can represent these transformations and so it benefited from larger filter patches. The translation approximation of the convnet is unable to properly align larger features to the true symmetries, and so it performed better with smaller filters. The convnet requires about four times as much data to reach the accuracy of the symnet with the smallest training set. Larger filters capture more structure than smaller ones, allowing symnets to generalize better than convnets, and effectively giving each symnet layer the power of more than one convnet layer.

The left side of the graph may be more indicative of the types of gains symnets may have over convnets in more realistic datasets that do not have thousands of images of identical 3D shapes. With the ability to apply more realistic transformations to sub-parts, symnets may also be better able to reuse substructure on datasets with many interrelated or fine-grained categories. Since symnets are a clean generalization of convnets, they should benefit from the learning, regularization, and efficiency techniques used by state-of-the-art networks [13].

# 7    Conclusion

Symmetry groups underlie the hardest challenges in computer vision. In this paper we introduced deep symmetry networks, the first deep architecture that can compute features over any symmetry group. It is a natural generalization of convolutional neural networks that uses kernel interpolation and transformation optimization to address the difficulties in representing high-dimensional feature maps. In experiments on two image datasets with 2D and 3D variability, affine symnets achieved higher accuracy than convnets while using significantly less data.

Directions for future work include extending to other symmetry groups (e.g., lighting, 3D space), modeling richer distortions, incorporating probabilistic inference, and scaling to larger datasets.

**Acknowledgments**
This research was partly funded by ARO grant W911NF-08-1-0242, ONR grants N00014-13-1-0720 and N00014-12-1-0312, and AFRL contract FA8750-13-2-0019. The views and conclusions contained in this document are those of the authors and should not be interpreted as necessarily representing the official policies, either expressed or implied, of ARO, ONR, AFRL, or the United States Government.

## Footnotes

[1]The neighborhood that defines a square filter in convnets is the reference point translated by up to k times in x and k times in y.

[2]Grid search over learning rate $\{.1, .2\}$, mini-batch size $\{10, 50, 100\}$, filter size $\{5, 10, 15, 20, 25\}$, number of filters $\{20, 50, 80\}$, pooling size (convnet) $\{2, 3, 4\}$, and number of control points (symnet) $\{5, 10, 20\}$.

[3]Grid search over filter size in each layer $\{6, 9\}$, pooling size in each layer (convnet) $\{2, 3, 4\}$, first layer control point translation spacing (symnet) $\{2, 3\}$, momentum $\{0, 0.5, 0.9\}$, others as in MNIST-rot.

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
