[Reviews · NeurIPS 2014]

Submitted by Assigned_Reviewer_6

This paper presents an extension of deep convolutional networks to general symmetry groups.
The authors present an algorithm to construct and learn feature maps over general transformation groups, such as the affine group, leading to a generalization of deep convolutional networks. The resulting network is evaluated using the affine group on Norb and Mnist-rotated, showing improvements in terms of sample complexity with respect to traditional CNNs.

This paper addresses an important limitation of current state-of-the-art object recognition architectures, namely the lack of prescribed invariance guarantees with respect to more general transformations beyond translations. Similarly as other works (which are appropriately cited), the authors propose a deep convolutional model, where convolutions are now over more general transformation groups. The main contribution of this paper is to introduce a supervised learning architecture with the ability to learn filters over the affine group.

Pros:
- The construction has the appropriate level of rigor and is mathematically sound.

- The paper is well written, and the figures are very helpful to understand the construction. As mentioned later, perhaps a toy analytical example where local invariance is shown through equations would help.

- The paper presents a novel variant of traditional CNNs. Although group invariance ideas are not new, the construction is novel and presents an interesting alternative to analytical constructions based on wavelets.

There are however some claims that are not completely justified and some issues relating to the complexity and scaling of the method.

- In the introduction, authors claim that "The discriminative training of symnets yields features with task-oriented invariance to their sub-features". I do not see how this can be achieved as long as the k-neighborhoods and the pooling scale, given by the covariance matrix Sigma, are shared across all feature maps.

- How robust is the method to downsampling? The authors introduce the notion of group alignment using warping to combat aliasing artifacts. I am slightly concerned with the complexity and stability of such approach. How many iterations are required per each control point (and per each feature map) ? How is the prior chosen?

- Local transformations. The paper describes that pooling layers achieve partial group invariance by performing local averages over the symmetry group. This is the key property of the model, which is illustrated in Figure 2. I think it would be very helpful to describe in further detail this local invariance mechanism with equations. That is, if x and x' are related via x = sum_p alpha_p y_p and x' = sum_p alpha T_{g_p} y_p, where y_p are 'parts' and T_{g_p} are transformed version of the parts, then show how the network achieves M x approx M x'; this would also allow us to see that invariance to local deformations is naturally achieved with deep architectures as opposed to shallow ones.

- Numerical experiments: Although the strengths of the paper are mainly on the model and theory side, I believe the authors should devote more effort on this section. I find the two datasets relevant for the present paper. Concerning the mnist-rot, the authors should perhaps mention that scattering networks achieve less than half the error rate (4.4% as reported in 'Invariant Scattering Convolutional Networks), and comment on the pros and cons of their construction with respect to steerable constructions such as scattering networks. Numerical performance on the two reported datasets is not 100% convincing, especially given the potential of the architecture. The fact that both curves meet for sufficiently large training set should be better analyzed. Is this because a traditional convnet can 'learn' the affine group (or at least the orbits present in the data) given enough data, or is it because the symnet performance might be limited due to the numerical approximation of control points and/or warping?

- An important question that the paper does not address is the question of Group Learning. How could one efficiently discover and use the most appropriate symmetry group given a particular dataset? In some sense, one can argue that local translations and their generalizations on higher layers of deep networks have the ability to model complex transformations acting locally in the data. Although such model of variability does not have the global properties of transformation groups, it might account for most of the useful intra-class variability in object recognition.

Summary: This paper presents an interesting and novel variant of Deep Convolutional Networks, with the capacity to build representations locally invariant to more general transformations.
The construction is well presented and numerical experiments are shown to improve over traditional CNNs, although further analysis and stronger experiments would make an even stronger claim towards its usefulness in object recognition deep architectures.

Submitted by Assigned_Reviewer_15

This is a very interesting and sensible paper. It has long been obvious that the trick of weight-sharing for features that are translated replicas could also be applied to other operations such as rotation and dilation. However, the obvious way of doing this leads to an explosion in the number of replicas of each feature that is exponential in the number of allowed types of operation. Itn is not until quite late in the paper that the authors explain their proposed solution to this problem: They perform local searches from multiple starting points for locally optimal fits of the multidimensional features. This is actually quite like the motivation for capsules in a paper they cite. Each capsule can only instantiate one instance of a complicated feature so I think there is a one-to-one correspondence between their starting points and convolutional capsules.

Apart from not explaining the lo9cal search idea much earlier (like in the abstract?) the paper is well-written and the results demonstrate that putting in more prior information via replication across full affine transforms leads to much better generalization on small data-sets. It is a pity they do not have results on ImageNet but the results they do have are fairly convincing.

Technically the paper seems to be very competent and I couldn't find any errors.
Summary: Use weight sharing across full affine transformations and instead of enumerating all possible replicated filters use local search from a number of starting points to find locally optimal fits of the filters that can then be interpolated.

Submitted by Assigned_Reviewer_39

This paper proposes to extend usual convolutional networks from the
translation symmetry group to the affine group (symnets). This entails
computation that may grow as k^6 instead of k^2 when each feature is
computed on a window of width k in each of the group dimensions. To deal
with this computational complexity, the paper proposes to optimize the
location (in R^6) of N control points located in the space of the group,
for each feature map. The output of the convolution+subsampling can then
be approximated by a form of weighted averaging between the control points,
and the whole thing can still be optimized by backprop + gradient-based
optimization.

This is a really interesting idea and it is also new (the part about using
optimized control points, not the part about extending convnets to
transformations other than translation) as well as attacking an age-old
problem, but the weak point of the paper are the experiments, along with
the selling tone used in many places.

The results suggest on the surface that symnets can generalize better on
very small datasets (a few hundred examples), but the appropriate control
experiments are not performed, and both the conclusions (and generally the
language of the paper) seem not in line with the standards of scientific
reporting, in an attempt to make the proposal and the results look better.

The experiments should have compared with a convnet that uses the same
prior information, i.e., with an augmented dataset, randomly transforming
along the affine group (which is the standard practice anyways). The
experiments should have compared networks of arbitrary depth and size
(optimized on the validation set), and not be restricted to networks of the
same size and depth, because it is likely that in order to emulate a 6-D
symnet, a regular convnet needs more resources (both in depth and number of
features and parameters), but these can be compensated by using an
augmented training set. Finally, a fair comparison should include
computation time (especially for training). One might suspect that the
proposed algorithm (which is not 100% clear, although the big picture is
clear) requires a lot more computation per training example than a regular
convnet. The reader needs to know how much more, in the specific
experiments reported.

A number of elements of that paper are not acceptable in the scientific
reporting standards: hiding the above computational cost information (of
which surely the authors are aware - not a single word is spoken about
training and test time), not doing the above proper and fair comparisons,
and then claiming that symnets generalize better without stating that the
advantage is mostly for tiny datasets with a few hundred examples. It is
unfortunate that many authors have a similar behavior, but still, it should
not be acceptable at NIPS.

Additional detailed points

Line 39: The authors write that convnets require enormous datasets. But there
are such datasets, so the question is whether one can do better with the given
datasets. It is true that we would like to generalize with few examples for
new classes for which there are not such large number of labeled examples.

The use of the word 'kernel' initially threw me off. Of course it is used here
in a sense different from the more common use in the ML community. I suggest
to clarify that early on.

Line 69: "we do not completely pool out a symmetry", but neither do convnets.
Similarly for discriminative training, same line. Clarify that you refer to
some other approach, maybe.

Line 241: "polynomial dependence on D", actually should be exponential in D,
and polynomial in k, i.e. simply k^D.

Line 290: why use the minimization of squared error rather than maximization
of the dot product? Is the norm of the feature vectors kept constant? how?

Line 303: "penalizing relative locations [10]". Not having read [10], the reader
would like to know briefly what this is about.

Figures 5, 6: it is crazy to compare only with a single-layer or a 2-layer convnet.
See main discussion above.

Section 6.2: what was N, at the end of the day?

Line 427: I found this conclusion really misleading "affine symnets achieved
higher accuracy than convnets while using significantly less data". The truth
is that when *only* a few hundred examples are available, and when the convnet
cannot exploit transformed augmented datasets and is highly limited in capacity
(all of which are not realistic), then the symnets generalize better. Clearly
this is a very different kind of message...

Line 431: what about the other hyper-parameters?

=========== POST-REBUTTAL NOTES

The authors have argued that "data augmentation is not feasible for many datasets" to justify the fact that they did not want to compare symmetry nets with convnets + data augmentation. The truth is that data augmentation has been for many years the standard way of training convnets in practice, it is cheap, simple, and effective, and helps to embody the same kind of prior information that symmetry nets try to exploit. Data augmentation does not need to visit the cross-product of all possible deformations: that's the beauty of it. A random sample of deformations along all the desired directions of invariance is sufficient to give a huge boost to convnets ESPECIALLY ON SMALL DATASETS like the ones the authors used. If symmnets can exploit that same prior information better than random deformations, then this is a very important piece of information!

The authors have also refused to change their misleading conclusion. That is sad. Most careful readers will not be fooled but will instead come out with a bad impression about the authors.

Nonetheless, I have raised my rating above the acceptance threshold because I believe I like the underlying idea, which is original and should be a good contribution to NIPS.

I strongly encourage the authors to reconsider the above two points in order to make their paper a more credible scientific contribution that will be more likely to have impact in the field.
Summary: Review summary: very interesting idea about using optimized control
points to extend convnets to the full affine transformation group,
but I am afraid that impact will remain limited until the authors do the proper comparisons
against convnets, i.e., by training the convnets with augmented data
with transformations, which is both easy and the norm in practice.
Author Feedback
Author rebuttal: We would like to thank the reviewers for their careful consideration and helpful suggestions.

REVIEWER 39

Experiments should have compared with a convnet that uses augmented data.
>Data augmentation is not feasible for many datasets and symmetry spaces. The datasets we use already finely sample their respective symmetry spaces such that our learning curves measure the amount of augmentation that would be required. The way in which the symnet locally pools affine space cannot be duplicated by augmentation of whole-image transforms.

Experiments should have compared networks of arbitrary depth and size because it is likely that a convnet needs more depth and parameters, but these can be compensated by using an augmented training set.
>Given enough units, augmented data, and training time, a single layer neural network can theoretically memorize any image dataset. The goal of our experiments is to show performance with comparable network size and finite data.

Computation time?
>We did not precisely measure times, but the symnet experiments in the paper were no more than 10x slower than those for convnets (we should add this). Convnets have been around for many years and have highly optimized CPU and GPU implementations. We think it is worth sharing this research prior to optimization.

The word 'kernel' is used differently from the more common use in the ML community.
>We use kernels in the same sense as in “kernel regression” and “kernel density estimation”, as opposed to their use in SVMs.

L.69: This sentence does not seem to accurately reflect convnets.
>This sentence does not refer to convnets; it immediately follows two sentences describing Scattering networks and M-theory.

L.241: This should be exponential in D, and polynomial in k, i.e. simply k^D.
>Our use of polynomial dependence on D is correct; there is no dimension “k” in our paper. Computing the Gaussian kernel, for example, in D-dimensional space involves a (1xD)(DxD)(Dx1) vector-matrix-vector multiplication, which is O(D^2). The kernel trick is precisely what allows us not to have an exponential dependence on D.

L.290: Why minimize squared error? Why not just maximize the dot product? How is the norm of the feature vectors kept constant?
>Squared error is used by many alignment algorithms; the dot product is trivially maximized by brighter patches. We do not enforce constant norm.

L.303: Not having read [10], the reader would like to know briefly what this is about.
>This is a recent vision paper with nearly a thousand citations and a state-of-the-art method for object detection.

Section 6.2: What was N?
>N=40 in the first layer and N=16 in the second layer. This should be made more clear.

LL.39&427: It is only when a few hundred examples are available, when the convnet cannot exploit augmented datasets and is highly limited in capacity that the symnets generalize better.
>The conclusion is true for the datasets we examined, which have tens of thousands of examples. Note that the number of examples, augmentation, and capacity required by convnets go up exponentially with the dimensionality of the symmetry space. In contrast, symnets can generalize over high-dimensional symmetry spaces using kernels. Thus we expect that symnets will generalize even better compared to convnets on more complex datasets like ImageNet.

L.431: What about the other hyperparameters?
>We should add that the other hyperparameters were varied as with MNIST-rot.

REVIEWER 6

How can symnets have “features with task-oriented invariance to their sub-features” if the k-neighborhoods and the pooling kernels are shared across all feature maps?
>Invariance in symnets and convnets is a function of both filter weights and pooling functions. The filter weights are discriminatively-learned. It is possible to use different pooling functions for different features (l.305).

How robust is the method to downsampling?
>Symnets are more robust to downsamping than convnets because control points can have subpixel alignments.

How many iterations are required?
>Four iterations per control point per feature map for MNIST-rot (l.351). Control points were not optimized for NORB (l.400).

How is the prior chosen?
>The prior was designed for each dataset, chosen by validation performance.

I think it would be very helpful to describe in further detail this local invariance mechanism with equations.
>We will do this.

The authors should mention that scattering networks achieve less than half the error rate on MNIST-rot.
>Yes, we should mention this. That scattering network has m_max=2, so it would be more fairly compared with a two-layer symnet.

The fact that both curves meet for sufficiently large training set should be better analyzed.
>These datasets are simple and dense enough that, given enough data points, convnets can approximate all the affine transformations using just translation, but that won’t happen in sparser, more complex ones. We don’t think the use of control points is the reason, because their number and initial location can be varied as desired.

How could one learn a symmetry group given a particular dataset? One can argue that local translations and their generalizations on higher layers have the ability to model complex transformations acting locally.
>Learning symmetry groups from data is a fascinating research direction. Translations can approximate complex transformations, but as we show this requires more data.

REVIEWER 15

The proposed solution to the explosion of feature replicas is introduced late.
>We mention in the abstract and introduction how we address the explosion of the number of replicas of each feature (by kernel-based interpolation).

Relation of control points to capsules?
>The output of a low-level capsule is limited to the regions of symmetry space that it has been trained on. Symnet control points can evaluate untrained regions of symmetry space.